# A Machine Learning Based Discharge Prediction of Cardiovascular Diseases Patients in Intensive Care Units

**DOI:** 10.3390/healthcare10060966

**Published:** 2022-05-24

**Authors:** Kaouter Karboub, Mohamed Tabaa

**Affiliations:** 1FRDISI, Hassan II University Casablanca, Casablanca 20000, Morocco; 2LRI-EAS, ENSEM, Hassan II University Casablanca, Casablanca 20000, Morocco; 3LGIPM, Lorraine University, 57000 Metz, France; 4LPRI, EMSI, Casablanca 23300, Morocco

**Keywords:** cardiovascular diseases, discharge, Electronic Health Records, intensive care units, machine learning

## Abstract

This paper targets a major challenge of how to effectively allocate medical resources in intensive care units (ICUs). We trained multiple regression models using the Medical Information Mart for Intensive Care III (MIMIC III) database recorded in the period between 2001 and 2012. The training and validation dataset included pneumonia, sepsis, congestive heart failure, hypotension, chest pain, coronary artery disease, fever, respiratory failure, acute coronary syndrome, shortness of breath, seizure and transient ischemic attack, and aortic stenosis patients’ recorded data. Then we tested the models on the unseen data of patients diagnosed with coronary artery disease, congestive heart failure or acute coronary syndrome. We included the admission characteristics, clinical prescriptions, physiological measurements, and discharge characteristics of those patients. We assessed the models’ performance using mean residuals and running times as metrics. We ran multiple experiments to study the data partition’s impact on the learning phase. The total running time of our best-evaluated model is 123,450.9 mS. The best model gives an average accuracy of 98%, highlighting the location of discharge, initial diagnosis, location of admission, drug therapy, length of stay and internal transfers as the most influencing patterns to decide a patient’s readiness for discharge.

## 1. Introduction

Factors such as blood pressure, high cholesterol levels and the adoption of bad habits including smoking and highly fat-saturated foods led to double the number of patients with cardiovascular diseases in the period between 2000 and 2019 compared to 1990, according to the American Hospital Association’s (AHA) report published in 2020 [1]. In 2017, the Kaiser Family Foundation Analysis of the Organization for Economic Co-operation and Development (OECD) [2], reported that the United States of America (USA), compared to other developed countries, has fewer medical resources (2.6 practicing physicians and 2.8 beds per 1000 population compared to 5.2 and 7.4 in Austria, 4.3 and 8 in Germany, per 1000 population, respectively). On the other hand, data published by the AHA in 2018 [3] indicates there are a total of 5256 registered community hospitals in the United States of which 2704 (more than 51%) deliver intensive care services with a total of 96,596 Intensive Care Unit (ICU) beds. In total, 68,558 of these beds are dedicated to adults (46,795 medical-surgical, 14,445 for cardiac care and 7318 for other ICU needs), 22,901 for neonatal care, and 5137 are pediatric ICU beds. Geographically, the distribution is mainly in metropolitan areas with 74% of ICU beds followed by 17% in micro-Politian areas and 9% in rural areas. Such disparities arise from a lack of study, biased data, or a misunderstanding of the healthcare ecosystem. In its 2018 report, the Health Systems for Prosperity and Solidary by the World Health Organization (WHO) [4] mentioned financial crisis, political choices, variations in epidemiology and social preferences or variations in efficiency as the main reasons why some countries would not invest in the healthcare system. As for the uncertain nature of hospitals’ ecosystems, static planning of these medical resources seems to be an inconvenient solution. In the literature, many studies have been attracted by the complexity of such issues. Thus, most of these studies put on the surface the importance of dynamic predictions when trying to be one step ahead of ecosystem changes [5,6,7,8].

In the course of solving resource allocation problems in uncertain environments, such as ICUs, researchers focused on two points: (1) the huge possibility of using increasingly leveraged clinical data captured from Electronic Health Records (EHR) systems. (2) The need to predict patient outcomes as a step toward an efficient decision-making tool.

In fact, severity score systems were developed to predict a patient’s outcome and to compare quality-of-care and stratification for clinical trials [9]. The development of scoring systems involves a complex combination of clinical acumen and advanced statistical techniques. These scoring systems must be rigorously assessed in terms of accuracy, reliability, and methodological rigor before being introduced into clinical practice [10]. Most of them were first derived from a database of patients and their various physiological measurements during their ICU stay. Used databases are retrospectively analyzed to find which of the selected variables are the most predictive for a chosen outcome. The testing of a model on such a validation cohort cannot be considered to represent independent validation. The sample size and randomization process used to select this cohort from the starting database makes it inevitable that the development and validation samples perform interchangeably. However, in an upgraded special article, a group of internationally recognized clinical experts suggested that severity of illness scores should not be considered as a transition condition from ICU to lower acuity care wards. Instead, they specified that such prioritization identifiers might be used to assess high-risk populations after discharge and not their readiness to be transferred [11]. In fact, a review of multiple studies revealed that cardiovascular patient health status was evaluated from four main angles: (1) their cardiovascular disease history, the demographic and socio-economic factors [12,13,14], etc. (2) Their health status before and after medical interventions [15,16]. (3) The main psychological and behavioral factors interfering with their medical condition [17,18,19]. (4) Using the patient’s health status as a predictor for future outcomes such as mortality [20,21]. Taking these factors combined, it seems that even studies that focus on scoring system types, case studies, and guidelines for using such risk assessment tools [22,23,24] can come to the conclusion that severity score systems cannot be used to predict patients’ discharge.

On the other hand, a new trend in learning methods or machine learning developed in and for technology and healthcare industries offers tremendous potential to enhance medical research and clinical care, especially as providers increasingly employ EHR.

There are many areas that can benefit from the application of machine learning techniques in the medical field, mainly diagnosis and outcome prediction.

In fact, many studies, such as [25], showed that MIMIC-III specific machine learning models using only 10 clinical variables outperformed nine commonly used severity scoring methods. However, other related studies [26,27] have also shown that machine learning models outperform severity scores in predicting in-hospital mortality. Furthermore, developing health system specific prediction models using machine learning enables continuous improvements of the model by including more training data (as more data becomes available), adding new clinical or laboratory variables to the model, or re-training the model using newly-developed machine learning algorithms.

Following this axis and trying to explore new ways to use machine learning other than simulation and modeling in medicine, we developed multiple machine learning models to predict patients’ readiness for discharge who are admitted to ICUs. In other words, when a patient is admitted to ICU, they can be transferred to a lower care unit or sent home after receiving the necessary treatment and staying in the hospital for a period. The goal here is to predict the length of stay in one and only one ward in ICU, any other transfer is considered a discharge. In fact, accurate discharge prediction will enable decision-makers (healthcare providers in most cases) to have a clearer vision of future actions and prevent subsequent readmissions. We made sure to eliminate redundant elements and normalized measures, as the MIMIC-III is collected using CareVue and MetaVision clinical information systems. Our approach, based on correctly imputed missing elements in our dataset, uses different algorithms to compare the performance of different configurations of these models. In fact, these models will predict patients that are more likely to be discharged among other patients. The performance of our model is assessed using a Root Mean Square Error (residual mean) with respect to related characteristics: selection of variables and weights, used variables (age, origin, chronic health status, physiology, and acute diagnosis) and size of the validation population. Such a study is meant to identify the impact of drug therapy, type and time of admission, and processed transfers from non-ICU wards and an ICU’s cardiology department on the patient’s readiness for discharge. Moreover, to discover new opportunities using a variety of machine learning techniques other than previously mentioned machine learning models, or discrete event simulations that have studied resource allocation problems [28], to provide the percentage of patients that meet the discharge criteria and features learned in the model’s training process. This might help physicians to prioritize inspections—to assess discharge—to specific patients. The deployment of such models will aid in the decision-making process of healthcare workers by improving the prediction of premature deaths, making medical decisions about high-risk patients more efficient, evaluating the effectiveness of new treatments, and detecting changes in clinical practices. The rest of the paper is organized into two main sections. The first represents a literature review of main studies in the same scope, followed by a context and mathematical description of patients’ flow, then by the implementation, results, and discussion of these results compared to the available state of the art literature. We end the paper with a conclusion.

## 2. Literature Review

Discharge planning is, by consensus, suffering from a lot of variability in the clinical decision-making processes. Most ICUs do not use written patient discharge guidelines. Clinicians have rather little secure evidence upon which to base any decision about discharge location. Such ambiguity can lead to poor management of patients, which can result in premature discharge and, subsequently, death or readmission. This has been a factor in the motivation to create critical care outreach teams and triage models to improve discharge outcomes.

In fact, many studies have tried to determine the factors that can predict the length of stay, or discharge in general, where patients are limited to a certain health condition or group of health conditions [29,30]. We found many models with varying degrees of data specification and accuracy. In the paper [31], hospital records were analyzed to determine if any factors could predict hospital Length Of Stay (LOS) and readmission after colorectal resection through linear regression. The data used in this study is a combination of databases from the National Cancer Registry (NCR) and the Hospital In-Patient Enquiry Scheme (HIPE), which contains records of patients in Ireland. STATA (statistical software for data science) was used to determine the best variables for logistic regression using a combination of likelihood ratio tests. For the LOS, it was determined that age, higher levels of co-morbidities, and marital status were associated with an increased LOS. Another study analyzed hospital records to identify the predictors of an increased LOS after Acute Exacerbation of Chronic Obstructive Pulmonary Disease (AECOP) [32]. A multivariate logistic regression model was created to assess the predictors of early discharge in a period longer than 11 days. The results from the aforementioned study show that being admitted between Thursday and Saturday, having high PaCO2, low serum albumin level, or having heart failure, diabetes, or stroke are the most important predictors of a very long LOS. The LOS of patients with cardiac problems was the focus of [33], which is one of the few that employs machine learning techniques. The patient data was retrieved over a five-year period from a hospital in Iran that specializes in treating and researching cardiovascular conditions. Thirty-six different attributes were included per row and three different models were run on the same data: decision tree, neural network, and support vector machine. Out of the three, the support vector machine approach outperformed the two other models, with the diagnosis ICD-9 (International Statistical Classification of Diseases ninth version) code (this provides an internationally standardized code per disease), the diastolic blood pressure (blood pressure in the arteries between heartbeats) and age being the three most prominent input variables (highest relative weight). In the paper [34], a retrospective review of a database was conducted to determine the predictive factors for hospital stay and mortality. An analysis was performed on a database from the Cleveland Clinic who had undergone noncardiac surgery within a five-year period along with measurements of Mean Arterial Pressure (MAP), Bispectrality Index (BIS), and Minimum Alveolar Concentration (MAC). Through logistic regression, it was found that a “triple-low” value of MAP, BIS, and MAC were strongly correlated with an extended LOS.

In fact, simulation has been extensively used to evaluate the impact of resource availability and the organizational settings on healthcare outcomes quality and the costs related to medical interventions and patients’ stay [35,36]. There are several methods of simulation but most commonly they are classified into four main categories: Monte Carlo [37,38], Discrete Event Simulation [39,40], System Dynamics [41,42], and Agent Based Simulation [43,44]. Some of the very important advantages of using these simulation techniques are: 1. The ability to perform “what if” analysis that evaluates the performance of the system in different scenarios, considering many types of input data and model parameters as well as identifying critical points related to the system’s bottlenecks [45]. While 2. is the ability to perform these scenarios in different time windows [46]. As a result, simulation is useful when a problem exhibits significant uncertainties that require stochastic analysis.

Indeed, several studies have been conducted on the application of simulation as an effective tool to improve processes in healthcare systems to minimize healthcare costs and increase the satisfaction of patients [47].

The main problems in the healthcare system that are addressed for emergency patients based on simulation and modeling knowledge are resource allocation, and patient flow problems, while for non-elective patients it is mostly scheduling and bed assignment [48,49,50,51]. Many studies have been conducted to optimize processes and patient flow in the healthcare systems [52,53,54,55,56]. The optimized patient flow is defined as a high patient throughput, low patient waiting times and short LOSs, while keeping staff utilization rates high and reducing staff idle time. The increasing cost of providing high-quality health care has made hospital administrators minimize resources while still striving to provide the service with the desirable quality. Many studies [57] find simulation modeling attractive since it can estimate the operational characteristics of a complex system as well as monitoring the results of changes in planning and resource allocation prior to implementation, which minimizes the financial risks for decision makers. According to the field of study of this thesis, the literature review of the Discrete Event Simulation (DES) in validation-simulation studies is classified into two categories including patient flow and resource allocation.

However, it does not seem possible to elaborate on a universal, conclusive procedure for matching the most suitable simulation technique to a specific problem.

On the other hand, there are relatively few studies on discharge prediction in a critical care setting: they are exclusively focused on discharge readiness [58,59] or they are designed to predict a specific discharge destination [60]. For example, [61] used demographic, ICU admission, and ICU clinical data measured during the first 24 h of ICU admission to develop a predictive algorithm for the early identification of ICU patients with a high probability of discharge to a long-term acute care hospital. The study found that their predictive algorithm can accurately predict the likelihood of patients’ discharges. In addition, [62] investigated the relationship between vitamin D status at ICU admission and the home or non-home discharge destination for critically ill surgical patients. They suggested that vitamin D levels may impact patient-oriented outcomes in ICU, and it might be a modifiable risk factor for the discharge destination.

Alternatively, the prediction of patients’ discharge from ICU can be expanded to focus on the characteristics of the patients admitted—many tools have been developed to support discharge planning. Mainly, these tools try to predict the likelihood of complications during hospitalization. Furthermore, these tools try to predict functional adverse outcomes, which can pose serious difficulties during the discharge process [63]. Most of these tools are appropriate for patients admitted for medical conditions, and the majority of those are condition-independent and can be widely applied.

For all these reasons, more data are needed about those factors already present before proper intervention and that is associated with a longer LOS or a discharge with the need for additional care, and that can be investigated in the early phases of the treatment trajectory. The aim of this study was to investigate those factors in a large sample of patients, in order to better understand what can be done to predict, as early as possible, which patients will need personalized and more demanding discharge planning, and possibly to suggest general items suitable for this prediction in general care departments for patients with cardiovascular diseases.

## 3. Context and Methods

The determination of appropriate medical resource distribution in healthcare facilities is a very challenging task as it must be coordinated at three different levels [64]. At the macro-allocation level, patterns of this distribution are drawn by legislation, government funding mandates and healthcare insurance plans. At the organizational level, policies, clinical practice guidelines and protocols decide how resources might be allocated to make maximum use of limited resources. However, at a micro-allocation level, it is a physician’s mission to decide whether a treatment or an investigation is in a patient’s best interest or not [65,66].

### 3.1. Mathematical Context

In every patient’s discharge, the bed’s occupancy distribution is evaluated. Having more visibility on when and how to discharge patients give hospitals’ policymakers and physicians in ICUs more flexibility and the ability to draw admission patterns, face admission peaks and manage general wards and lower medical care units more effectively. Moreover, and more importantly, to deliver medical care to the maximum number of patients by reducing the LOS and increasing admission rates. Figure 1 represents the generic guiding flow map divided into two main units: admission flows and the ICUs’ In and Out flows. The model presented in Figure 1 has been evaluated and validated using Non-Homogenous Discrete Time Markovian Chains [67].

In ordinary circumstances, a patient in a critical condition might be admitted directly from the emergency room, be a planned admission, or from internal wards in which he or she was admitted as a non-critical case, then needed intensive medical care.

We have developed a model that aims to provide a generic representation of a hospital’s internal flows [68]. The aforementioned is based on discrete Markov chains and is validated using real-world data. The following is, therefore, intended to prove how it is important to optimize the number of inspections using machine learning techniques. This goes along with shortening the LOS but also to take into consideration the patient’s condition. In a more illustrative image, the mathematical model we are representing here will guide the usage of our dataset. This means that, at every step of the flow, we will take parameters that might impact the patient’s readiness for discharge at the end of their treatment.

Let $\chi_{k,j,t}\left(t\right)$ be the number of admitted patients of pathology $k=\left\{coronary\;artery\;disease,\;congestive\;heart\;failure,\;acute\;coronary\;syndrome\right\}$ admitted as $j=\left\{Emergency,\;Elective,internal\;ward\right\}$ department after spending t time in the hospital. $O_{k,j,t}$ represents the overflowing patients from both the external and internal admissions of previously named pathologies. This overflow is calculated considering the patients in hospital and still not served, and scheduled patients not arriving in time. In our model, we consider that all causes leading to overflow can result in very long diagnosis times.

At a given time window, we consider $W\left\{1\ldots q\right\}$ as the number of ICUs wards. We define the bed occupancy function as representing if a ward q is occupied by a patient of pathology or condition type k.
(1)$$U_{q,k}=\frac{number\;of\;patients\;of\;pathology\;k\;allocated\;to\;ward\;q}{total\;number\;of\;beds\;in\;ward\;q}$$

Let $F_{k}={𝓌}_{k}-\underset{k}{\sum }U_{q,k}$ be the number of free beds in ward *q*. The $D_{q,k}\left(.\right)$ reflects the bed’s distribution in a given time interval and ${𝓌}_{k}$ is the number of beds in ward k.
(2)$$D_{q,k}=\left(\left(\begin{array}{ccc} U_{11} & \cdots & U_{1M}\\ \vdots & \ddots & \vdots \\ U_{M1} & \cdots & U_{MM} \end{array}\right),\left(F_{1},F_{2},\ldots ,\;F_{M}\right)\right)$$
where $\left(U_{11},U_{22},\ldots ,\;U_{MM}|k=q\right)$ is the number of patients with primary hospitalizations.

We also define $\alpha_{k}$ and $\beta_{q,k}$ as the primary and secondary hospitalizations’ rate represented in Equations (3) and (4), respectively:(3)$$\alpha_{k}\equiv \alpha_{q,k}=\frac{\sum_{q,k\left(k=q\right)}U_{q,k}}{\sum_{q,k}D_{q,k}}\;$$
(4)$$\beta_{q,k}=\frac{\sum D_{q,k}-\sum U_{q,k}-\sum_{q}F_{q}}{\sum_{q,k}D_{q,k}}$$

Based on the work presented in [43], we can conclude that the distribution of the newly arriving patient to the different wards follows the process shown below:(5)$$\left\{\begin{array}{c} \chi_{k,j,t}*\alpha_{k}\;if\;F_{k}>0\;for\;k\in M\;\\ \chi_{k,j,t}*\beta_{q,k}\;if\;F_{k}=0\;and\;F_{q\neq k}>0\;for\;q\in M \end{array}\right.$$

In the matrix above, a newly arriving patient is allocated to the preferred ward in cases where there exists a free bed. Otherwise, the patient is oriented to another ward that may serve a similar service as shown in Figure 1.

We define in the following, that the time spent by a patient of type k as an inpatient in ward q is denoted $\tau_{q,k}^{k}$.

We also define $Q_{q,k}^{n}$ as the number of patients of type k in ward q right after the $n^{th}$ inspection. We can assume that the inspection and discharge patterns can be described by the same distribution and are the same from an operational perspective. The dynamics of such distribution can be cited as follows:(6)$$\left\{\begin{array}{c} Q_{q,k}^{n+1}\left(\alpha_{k}=1\right)=Q_{q,k}^{n}+(1-\sum_{k\epsilon M}\beta_{q,k})*{\left(\chi_{k,j,t}\right)}^{n+1})-\xi_{q}^{n+1}for\;every\;k\\ Q_{q,k}^{n+1}\left(\alpha_{k}<1\right)=Q_{q,k}^{n}+\left(1-\alpha_{k}\right)*{\left(\chi_{k,j,t}\right)}^{n+1}-\xi_{k}^{n+1} \end{array}\right.$$
where $\xi_{q}^{n+1}$ is the number of patients discharged in the (n + 1)th inspection and
(7)$$\Gamma^{N}=\sum_{n=0,\;\forall q,\forall k}^{N}\xi_{q}^{n+1}$$

In the same way, a decision to discharge a patient takes into account the patient’s pathology, condition and the period already spent as an inpatient. The main goal of expressing the number of inspections per period is to relate this factor with the service rate and occupancy function. Thus, in used data, we will have to divide the 24 h into specific intervals and set a mean number of inspections. This will help derive a discharge rate per period, which will provide visibility to how many primary and secondary hospitalizations are made and how many patients to admit.

### 3.2. Data Description

In our study, we used the MIMIC-III, which contains de-identified health-related data of more than 40,000 patients with more than 50,000 hospitalizations in the ICUs of the Beth Israel Deaconess Medical Center in Boston, Massachusetts, in the period between 2001 and 2012. The data in the MIMIC-III is divided into 26 tables, each one comprises a specific type and flow of data such as demographics, vital signs measurements, laboratory test results, procedures, medications, caregiver notes, imaging reports and discharge mortality.

The tables are linked by primary identifiers such as subject-ID, hadm-ID and ICUStay-ID. The recorded measurements are provided by Philips and iMDSoft tools used in CareVue and MetaVision clinical information systems, respectively [69].

The de-identification of patients was incorporated in accordance with the Health Insurance Portability and Accountability Act (HIPAA) standards and the federal code of the USA. It included the removal of the eighteen identification data elements defined by the HIPAA such as patient name, phone number, address, and dates such as the date of birth, date of admission, etc. In the following, we calculated the age of the patients and used their associated dates in an intervals-based approach by which dates are shifted into the future, between 2100 and 2200, in a manner to preserve the intervals [70,71].

For reasons of having the highest mortality rates [72,73] and being responsible for the utilization of a significant proportion of healthcare resources [74], we managed to solely use data related to adult patients with specific types of pathologies as shown in Figure 2. The current study basically includes 4402 admissions and a total of 4226 patients, as some patients are admitted more than once with different admission identifiers. In total, 2804 of them were admitted to the emergency room, 1466 were elective patients, and 132 were admitted as urgent. Of the total number, 2808 were diagnosed with coronary artery disease, 1315 with congestive heart failure, and 279 with acute coronary syndrome while admitted to ICUs and before receiving any prior treatments. Table 1, below, resumes the patients’ related data, input characteristics, outcome characteristics and actual measurements performed.

### 3.3. Preprocessing

Coronary artery disease is a condition where the coronary arteries are narrowed or blocked causing chest pain, congestive heart failure or/and an acute coronary syndrome condition. Such a blockage in the blood supply to the heart muscle area, or ischemia, can cause heart tissues to die within a few minutes. To relieve pain by reducing the heart’s workload, and to prevent chest pain and acute coronary symptoms from happening, doctors usually use nitrates, beta blockers, calcium channel blockers, ACE inhibitors, statins, antiplatelet and anticoagulants drugs, respectively (and sometimes combined) to reverse coronary artery narrowing or to open a blocked artery. Although, on the one hand, heart failure is a disorder related to the heart’s inability to follow the body’s demands in terms of blood flow, congestion of blood and regular beating; this condition is often caused by cardiac causes such as coronary artery disease, myocarditis, heart valve disorders, as well as some non-cardiac causes such as high blood pressure, anemia, kidney failure and others. The treatment of such a condition may, thus, include diuretics and nitrates to relieve symptoms related to the pain such as angiotensin, ACE inhibitors, beta blockers and aldosterone antagonists to help the treatment to succeed.

In the process of understanding the physiological parameters of patients, we noticed that some of these parameters are recorded at a lower frequency compared to others, such as non-invasive blood pressure. Such missing data we could handle using invasive blood pressure, which was continually monitored. In other cases, missing data were replaced by the median of the concerned variable such as the LOS. We also extracted informative patterns from patients’ physiological data and the chronological order of events between admission and discharge. Those features were first identified from MIMIC-III admission, chart events, ICU stays and prescription tables based on their known logistic and clinical relevance to our target endpoints. We determined the univariate importance of each feature with respect to the target variable. Such a technique enabled us to make our dataset easier to interpret and to understand how data is distributed within this specific population of patients as it addresses the interrelations between the different timing and physiological features and discharge decisions. Figure 3 represents the feature selection based correlation of chronological features related to the time when patients were admitted and discharged from ICUs, in addition to the importance of the features related to prescriptions and physiological patterns.

As shown in Figure 3a,b, the use of metoprolol, vancomycin, 0.9% sodium chloride, insulin, heparin, 5% dextrose (noting that it is also abbreviated as D5W in the MIMIC-III database), iso-osmotic dextrose, ondansetron ODT, phenytoin, piperacillin-Tazobactam-NA are used to prevent heart attacks by lowering blood pressure, preventing the formation of blood clots, and fighting bacterial infections. Although a constant rate of admissions to the emergency department and surgery is more likely to be followed by a peak in discharges, as might be noticed, admission and discharge from ICUs follow the patterns of global admission—the more admissions increased, the more ICU admissions and discharges increased and decreased, respectively (Figure 3c,d).

On the other hand, the starting and ending dates are more likely to be constant, which means that patients of similar pathology take similar durations under a given description. Drugs such as cholesterol lowering medications, insulins, ACE inhibitors, diuretics, beta blockers, glucose elevating agents and calcium channel blockers and their combinations designed, in most cases, for arrhythmia, artery disease, coronary disease, chest pain and hypotension have the highest patients’ early discharge probability. In the dataset files, there is no indication of an undesirable event experienced by a patient related to drug therapy that interferes with achieving the desired goal of such therapy.

## 4. Results and Implementation

Severity of illness is a composite of the magnitude of the acute disease, the patient’s physiological reserve, and the concurrent level of treatment and organ system support. Of these three variables, the physiological reserve is the most difficult to quantify and modify.

It is generally assessed using functional capacity, co-morbid disease, and age. Loss of functional capacity is an important predictor of frequent hospitalizations and death, and co-morbid disease impacts ICU and hospital outcomes [75].

In our study, feature importance reflects how useful and valuable every single attribute was when building the decision to discharge a patient. This importance is calculated for each attribute in our dataset, by which all attributes are ranked and compared to each other. Every single importance index is calculated by the average training loss reduction gained when using a feature for splitting. As noticed in Figure 3, the discharge location, ICU admission related diagnosis, admission location, drug, out-time (at which patients are transferred from the ICU unit to a recovery ward or to home), frequency of the given drugs, ICU stay ID (which reflects if a patient has an ICU admission history), start date (the date on which drug therapy started), LOS (length of stay), end date (the day in which the drug therapy ended), subject ID (which reflects, if associated with different admission ID, a patient’s readmission), the admission date, and the times by which the patient is admitted to ICU are the most informative features. As a consequence, we are using these features as input for our models. Missing values of numerical data are imputed using a basic algorithm of decision trees, while categorical data are encoded and linearly regularized. We experimented with eight models to identify the most suitable one. We have also set the decision tree’s regression as the baseline to estimate performance. Further, three of these models: 1. AVG blender; 2. Advanced Generalized Linear Regression Model; and 3. Efficient Neural Network, are ensembles of other models. Due to their iterative nature, Gradient Boosted models are almost guaranteed to overfit the training data, given enough iterations. Table 2 represents the tuning parameters of the baseline models, while all the other models are sophisticated forms of them. Any other parameters not mentioned in the table are left in their default values.

Figure 4 details the dynamics of the different algorithms used in our approach:

The overall workflow used for building the predictive models in the present work is illustrated in Figure 4. We have used Python 3.9 as a programming language to build our models. We used Pandas [76], NumPy [77], SciPy [78], TensorFlow [79], and time [80]. To ensure that our results are not biased towards a specific learning algorithm and to minimize the risk of over-fitting, we have implemented a set of machine learning predictors. To achieve the best performance, it was necessary to find the best hyper-parameters for each algorithm. A grid-search was conducted to find the best parameters for each model. A grid-search combines all possible combinations in a parameter grid where one defines the possible values for each hyper-parameter. In other words, it provides an exhaustive method to evaluate the combinations of all hyper-parameters.

The best possible parameter settings found using grid-search for each algorithm are presented in Table 2:

These methods were used to predict the discharges in a week. To study the impact of the data partition on the models’ performance, we conducted four experiments on different data partition configurations. The configurations and results after simulations are summarized in Table 3.

The model achieved an optimized training accuracy value of 0.98 and a mean residual = 0.0004 when all selected features and an advanced generalized linear regression model were used. In Figure 5, below, the discharge location, diagnosis, admission location, drug, out-time (day of the month), prod-strength, day (in a month) in which drug therapy starts, length of stay, (day in the month) in which drug therapy ends, admission time, admitted for the first time or not, time (in the day) in which a patient is admitted to an ICU, and the time in which the patient is supposed to be discharged to a general ward are the most relevant features, with 99.97%, 88%, 74%, 72%, 68%, 65%, 40%, 39%, 38%, 30%, 28%, 28% and 23% of the weights associated to all attributes in accordance with their importance in the training phase, respectively. This approach works by monitoring the performance of our models and automatically selecting the inflection point where performance on the test dataset starts to decrease while performance on the training dataset continues to improve. 

As mentioned previously, the figure above represents spots or prediction areas within which a given discharge readiness estimation is given. The spots are estimated using the Regressor Fit algorithm. Patients readmitted to emergency within 6 months of their first admission, patients with respiratory failure, and patients with low rates of multi-vitamins are the patients more likely to stay longer in hospital. In addition to this, the discharge location—home, or to a long-term care facility—is a key predictor in deciding if a patient can be discharged or not. Figure 6 represents the mean residuals obtained for each model.

In general, mixed or ensemble algorithms outperformed the other models. Firstly, in terms of performance, where advanced generalized linear regression, efficient neural network, and AVG blenders showed an accuracy of more than 90%. Secondly, in terms of robustness, where in multiple data partitions, these models also showed great results. The recapitulation of these results is presented in Table 3.

It can be seen in Figure 5 that by delaying the prediction somewhat into the admission time, a better prediction can be made. This result is in line with the hypothesis that complementing the data from the emergency service with the data collected at the ward would increase performance. Important features, such as suspected diagnosis and planned transfer to other wards, are added at this stage and this type of specific information can be valuable, especially when combined with all of the lab parameter values. For the admission stage, the balanced probability to be discharged ranges from 0.6 to 0.62. These results were in line with the ones obtained in previous LOS studies. The fact that the patient group used in this dataset was quite broad shows that there is a potential for these machine learning algorithms to be used in more generalized settings in hospitals. Patients using certain types of drugs, such as Hydralazine are more likely to be discharged after 5 days of admission. Earlier research has usually very specific limitations on patient type, such as diabetes or brain surgery patients, while this project focused on a certain hospital clinic and considers interference from outside factors. One of the limiting factors was low precision for the long-staying class (i.e., patient contacts staying longer than 3 days). At the discharge stage, this value ranged from 0.4 to 0.48. Such low precision would make it troublesome to use the prediction in a real system.

## 5. Discussion

In general, most discharge guideline reports published by the WHO [81], the Society of Critical Care Medicine (SCCM) [82] or the AHA [83], published between 2016 and 2018, list the following discharge criteria for ICU patients: stable hemodynamic parameters, stable respiratory status and airway patency, oxygen requirements not more than 60%, intravenous inotropic support and vasodilators are no longer necessary, cardiac dysrhythmias are controlled, neurologic stability with control of seizures, patients who require chronic mechanical ventilation resolved, and patients with tracheostomies who no longer require frequent suctioning. Discharge planning is multi-factorial and a succession of consecutive parts. One of these parts is the patient’s readiness for discharge. The assessment of such treatment transition enables a provider to estimate the patient’s and family’s ability to leave the original medical care institution and to move to a home health phase or to a lower intensity care area. A readiness for discharge from an ICU assessment requires the evaluation of the patient’s physiological stability, cognitive and psychomotor ability to carry out self-management regimens, social support availability and permanent access to healthcare systems [84]. Assessing the patient’s readiness for discharge from an ICU is a necessary task for the patient’s care and the equitable usage of the ICU’s available resources. In both cases, for ICU-Emergency admitted patients or ICU-Elective patients (postoperative monitoring and medical interventions), the determination of discharge line between intensive care and recovery care may withdraw occupancy rates of ICU beds. Discharge criteria enable a triage-based decision; allowing patients to leave ICUs if not necessarily needed, and by increase the rate of readmission.

Both admission and discharge involve a change of location with the potential for gaps in communication that may result in diminished and discontinuous care. There is growing research evidence showing that the outcomes of intensive care are affected by the timing of admission and discharge decisions, which in turn, are influenced by resource availability in the ICU and probable inexpert care on the ordinary wards [85]. Admission to the ICU from 00:00–07:00 h, and at weekends is associated with a higher mortality, as is discharge from the ICU to the ordinary ward at night [86]. Readmission to intensive care is associated with a hospital death rate 2–10 times that of non-readmitted patients [87] and can be mitigated by intensive care outreach in the form of intensivist-led rapid response teams [88]. Of the high-risk surgical patients admitted to intensive care in 28 European countries, 43% of deaths occurred after discharge to the ordinary ward [89], which suggests there are potential opportunities to look at again with regards to the way in which a patient’s readiness for discharge is assessed. Unintentional discontinuation of chronic medications is also common following discharge from the ICU and is associated with adverse patient outcomes. Decisions to admit patients to ICU or to discharge them to the ward are determined by the severity of their illness.

Our approach can be used in two different pathways. Firstly, to identify patients who are most likely to be discharged in each day per week and give them an uncertainty rate. Following these recommendations, hospital staff can prioritize these patients to be inspected, and then discharge them as early as possible so that other patients can be admitted to ICUs. Secondly, the possibility of ranking patients into mild and moderate severity while discharged. This allows the hospital to prioritize who and when to run the remaining tasks for patients on the discharge list. Despite the many advantages of our approach, it is important to highlight that the performance of our models may improve with a larger dataset in terms of accuracy but may exponentially decrease in terms of execution time.

A range of different tools and methods have previously been proposed, as shown in the comparative Table 4, with the aim of improving ICU discharge practice. These tools range from criteria for evaluating discharge readiness [90,91], to guidelines for discharge planning and education [92].

Predictive models based on very large patient numbers capture more population information than the individual clinician can acquire in a lifetime; however, the clinician will know more about the individual patient than any of these models can. For this reason, predictive systems may inform clinical judgement, but cannot replace it. Triage protocols to maximize the use of scarce resources in high seasonality periods have been modelled prospectively and retrospectively [93], demonstrating theoretical value in releasing intensive care beds by denying admission to those categorized as being too well or too sick to benefit. Several models have been developed to inform safe and timely ICU discharge decisions. Simple univariate risk factors include prolonged LOS, unstable vital signs (including tachypnea or tachycardia) and poor pulmonary function. For example, [94] have modelled post-ICU mortality and ICU readmission using data from more than 700,000 patients, incorporating admission diagnosis, severity of illness, laboratory values, and physiological variables in the last 24 h of the ICU stay. The Stability and Workload Index for Transfer score [95], and a model developed in France [96], have similar predictive precision for ICU readmission. Others have identified the potential for important reductions in mortality had triage models been used to avoid premature ICU discharge.

**Table 4 healthcare-10-00966-t004:** Results and Benchmarking.

Ref	Methods and Approach	Dataset	Metrics and Results	Scoring of Recommendation Strength
[97]	Focus: prognostication of clinical outcomes in ICUs. Methods: multivariate imputation by chained equations for missing data imputation. Adaboost, parRF (parallel implementation of a random forest), SVMRadialWeights (SVM with radial basis function kernel and class weights), avNNet (averaged Neural Network) and deep NN as classifiers.	Critical Care Health Informatics Collaborative (CCHIC) data infrastructure (22,514 intensive care admissions of which 21,911 were used in the study; 90.8% of them were alive at discharge.)	On day 2 (AUC): parRF: 0.853 simple and 0.857 cumulative. avNNet: 0.864 simple and 0.879 cumulative. Adaboost: 0.862 simple and 0.879 cumulative. svmRadialWeights: 0.849 simple and 0.884 cumulative. DeepNN: 0.881 simple and 0.895 cumulative	Larger data improves model’s performance.
[98]	Focus: mortality prediction, LOS prediction and ICD-9 code group prediction. Methods: SAPS-II (Simplified Acute Physiology Score) and SOFA. Super learner models, RNN and FFN.	Medical Information Mart for Intensive Care III (MIMIC-III) (v1.4)	SuperLearner-I: AUROC = 0.8448 and AUPRC = 0.4351. SuperLearner-II: AUROC = 0.8701 and AUPRC = 0.4991. FFN: AUROC = 0.8496 and AUPRC = 0.4632. RNN: AUROC = 0.8544 and AUPRC = 0.4519. MMDL: AUROC = 0.8664 and AUPRC = 0.4776. Scoring methods: 0.8035 AND 0.7322 for AUROC (SAPS-II and SOFA respectively), AUPRC: 0.3586 for SAPS-II and 0.3191 for SOFA.	Larger data improves model’s performance.
[99]	Focus: prediction of final diagnosis and clinical outcomes. Methods: universal language model fine-tuning for text classification (ULMFiT)	Medical Information Mart for Intensive Care III (MIMIC-III)	Accuracy: 80.3% for diagnosis top10, 80.5% procedure top10, 70.7% diagnosis top50, 63.9% procedures top50.	Larger data improves model’s performance.
[100]	Focus: in-hospital mortality prediction. Methods: deep learning networks.	Medical Information Mart for Intensive Care III (MIMIC-III) (42,818 hospital admissions of 35,348 patients)	Mortality prediction: AUROC: 0.9178 with data of all sources (AS) and 0.9029 with chart data (CD). PRAUC: 0.6251 for AS, 0.5701 for CD. LOS prediction: AUROC: 0.8806 for AS and 0.8642 for CD. PRAUC of 0.6821 and 0.6575, respectively with AS and CD.	Larger data improves model’s performance.
[101]	Focus: ICUs discharge prediction. Methods: random forest (RF) and logistic classifier (LC).	Bristol Royal Infirmary general intensive care unit (GICU) (1870 intensive care patients) and 7592 from MIMIC-III.	On the MIMIC dataset: AUROC(RF):0.8859, AUROC(LC): 0.8726. Accuracy (RF): 0.8531 and accuracy (LC): 0.8494. sensitivity (RF): 0.9049 and sensitivity (LC) is 0.9001.	Larger data improves model’s performance.
[102]	Focus: prediction of discharge location in ICUs. Methods: National Early Warning Score (NEWS/NEWS 2)	Surgical, coronary, cardiac surgery recovery, medical and trauma surgical intensive care patients with single admission in ICUs in a US hospital.	The NEWS AUROC (95% CI): all patients 0.727 (0.709–0.745); Coronary Care Unit (CCU) 0.829 (0.821–0.837); Cardiac Surgery Recovery Unit (CSRU) 0.844 (0.838–0.850); Medical Intensive Care Unit (MICU) 0.778 (0.767–0.791); Surgical Intensive Care Unit (SICU) 0.775 (0.762–0.788); Trauma Surgical Intensive Care Unit (TSICU) 0.765 (0.751–0.773).	Larger data improves model’s performance.
[103]	Focus: risk scoring in ICUs. Methods: attentive deep Markov model (AttDMM).	MIMIC-III with 53,423 ICU stays.	AttDMM with AUROC of 0.876.	Not specified
[104]	Focus: ICU readmission prediction after 24 to 72 h of discharge. Methods: fuzzy modeling and tree search feature selection technique.	MIMIC II (data of 4 different ICUs 26,655 patients, of which 19,075 are adults; 38% of the adult patients stayed at the medical ICU (MICU), 27% at the surgical ICU (SICU), 20% at the cardiac surgery recovery unit (CSRU) and 15% at the critical care unit (CCU))	AUROC of 0.76 and *p*-value of 0.006 with sequential forward selection. AUROC of 0.68 and *p*-value under 0.05 with sequential backward elimination.	Not specified
[105]	Focus: length of stay prediction in ICUs. Methods: neural network (random forest)	MIMIC-III (31,018 chosen data points)	Accuracy of 80%.	Larger dataset might improve the model’s performance.
Our model	Preprocessing: Regularized Linear Processing, Ordinal encoding of categorical variables, Tree based Algorithm. Prediction tools: Auto-tuned stochastic Gradient Descent Regression, Decision Tree, Extreme Gradient Boosted Trees Regression with Early Stopping, Gradient Boosted Greedy Trees Regression with Early Stopping, Light Gradient Boosted Trees Regression with Early Stopping, Advanced Generalized Linear Regression Model (GLRM), Efficient Neural Network (ENET), AVG Blender	MIMIC-III	{18% ^1^, 36% ^2^, 72% ^3^}: Accuracy = 98%, RESIDUAL MEAN = 0.000001, Prediction time: 123,450.9 mS. {18% ^1^, 36% ^2^, and 80% ^3^}: Accuracy: 97.8%, RESIDUAL MEAN = 0.000001, Prediction time: 123,551mS. {18% ^1^, 46% ^2^, 72% ^3^}: Accuracy: 93.7%, RESIDUAL MEAN = 0.000319, Prediction time: 215,693 mS. {25% ^1^, 36% ^2^, 72% ^3^}: Accuracy = 96.5%, RESIDUAL MEAN = 0.00147 and Prediction time: 15,039.3 mS. ^1^: Validation, ^2^: Cross validation, ^3^: Holdout	Larger dataset improve model’s performance

## 6. Conclusions and Perspectives

To optimize hospitals’ resources and maximize service quality, we presented a method to handle CVD patients’ discharge from ICUs using their physiological data and history of treatments, and on the learning capacity of ML algorithms. Firstly, we included physiological, hospital internal transfers associated with patients and prescriptions in their treatments from the MIMIC-III ICUs database recorded in the period between 2001 and 2012. The database contains de-identified, health-related data of more than 40,000 patients with more than 50,000 hospitalizations. The used dataset includes pneumonia, sepsis, congestive heart failure, hypotension, chest pain, coronary artery disease, fever, respiratory failure, acute coronary syndrome, shortness of breath, seizure, transient ischemic attack, and aortic stenosis that have the same ICU admission, flow mutation and prescription patterns. Multiple ML algorithms were used to compare the effectiveness of each model. We ran multiple experiences on the processed dataset to study the importance of the dataset’s size in every learning phase of our models. We tested our models using data related to 4226 cardiovascular disease patients. As a result, we achieved a better accuracy performance of 0.98 and an RM (Residual Mean) = 0.0004 using advanced generalized linear models, which included stochastic and trees regression. These results encourage future work that will include studying the impact of such a decision support tool on internal logistics and post-discharge outcomes. All methods need prospective validation.

Furthermore, the dataset available in this project contained a lot of information that was not included in the training and testing data due to the amount of feature engineering it would have required. As the whole pipeline, from raw data to polished flat-table format, had to be implemented, not all possible information could be extracted due to time limitations. This is one of the most interesting points to explore in similar projects in the future. The dataset contained copious amounts of time-series data from different lab tests, and it would be very interesting to develop more features connected to trends present in these types of data. Only statical features were used in this project, but if features could be extracted that reflect how, for example, a vital parameter has varied over time, this could prove very valuable to the ML models as trends are very important when clinicians evaluate patients. It would, therefore, be interesting to explore the feature engineering aspects of this dataset more extensively, not only for the time-series data.

## Figures and Tables

**Figure 1 healthcare-10-00966-f001:**
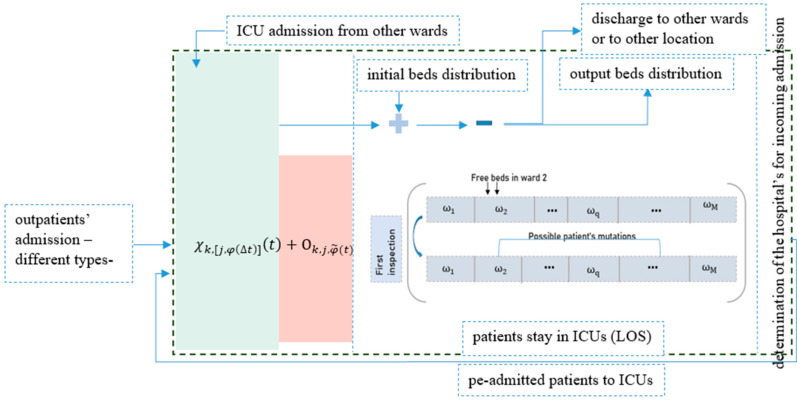
Flow dynamic in ICUs.

**Figure 2 healthcare-10-00966-f002:**
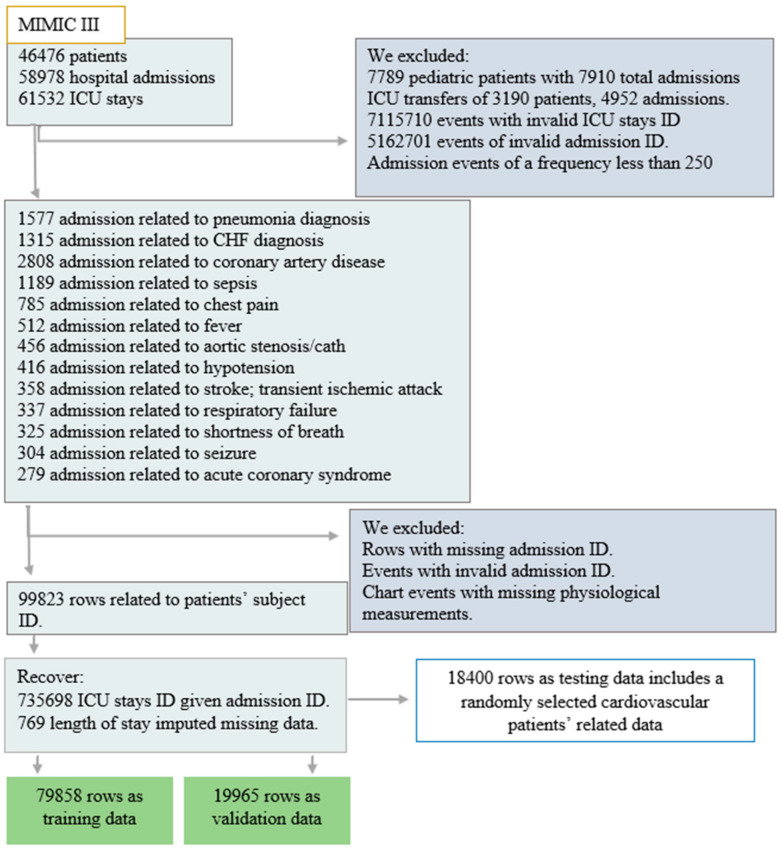
MIMIC III dataset generation process.

**Figure 3 healthcare-10-00966-f003:**
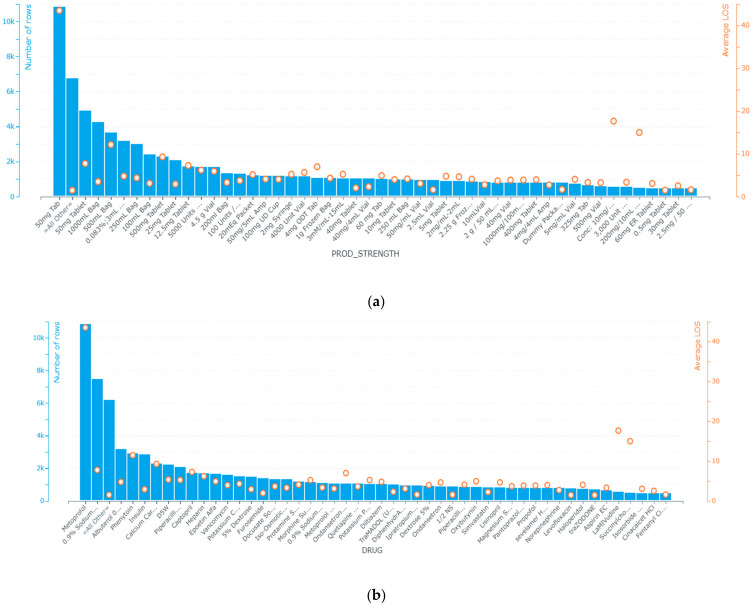
Feature selection of informative features. (**a**,**b**) represent the correlation between drug dosage and type, and time of discharge, respectively. Figure (**c**) is the time that marks the end of a specified drug therapy. (**d**,**e**) are the chronological timing of admission to ICUs, and their actual discharge time.

**Figure 4 healthcare-10-00966-f004:**
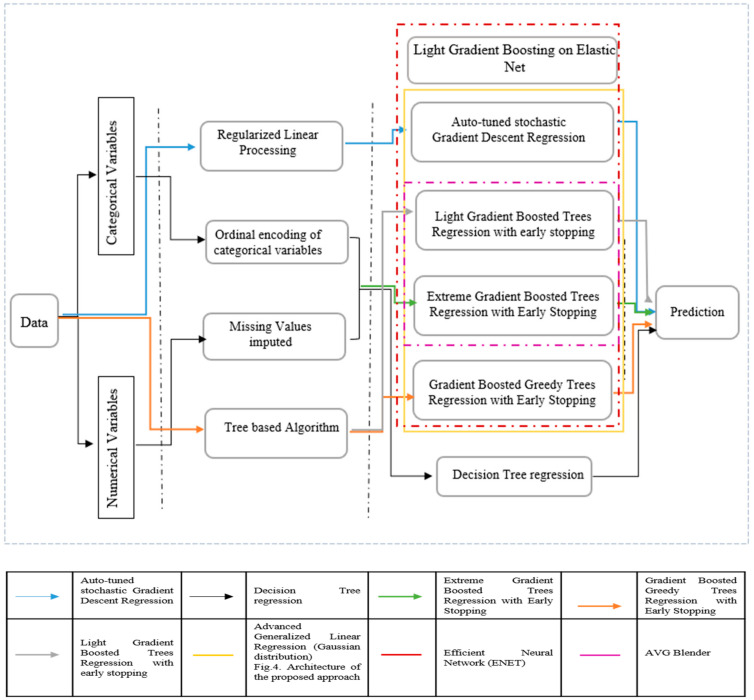
Architecture of the proposed approach.

**Figure 5 healthcare-10-00966-f005:**
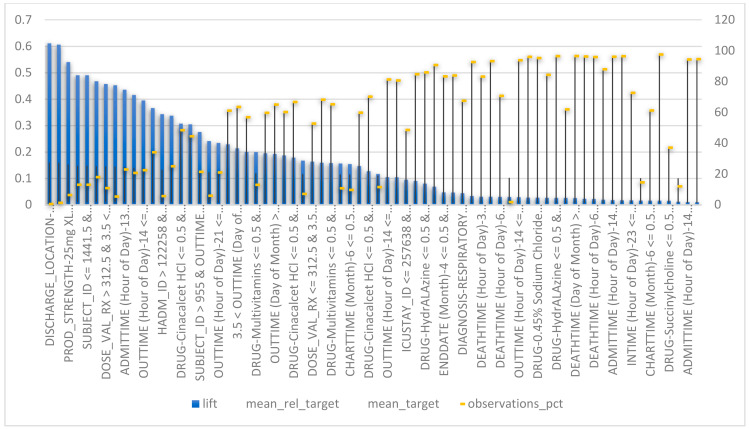
MIMIC III: training results.

**Figure 6 healthcare-10-00966-f006:**
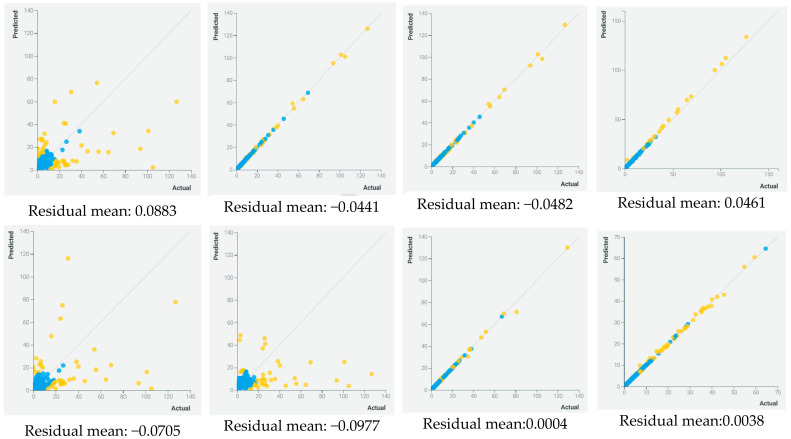
Residuals of different models (yellow represent expected outputs and blue ground-truth output).

**Table 1 healthcare-10-00966-t001:** Baseline patients’ characteristics and outcome measures.

	Overall Population Characteristics	Dead at Discharge Characteristics	Alive at Discharge Characteristics
*Average age*	65	73	64
*Sex (% men)*	53%	54%	57%
*Admission type:*			
Emergency	2804	226	2577
Elective	1466	14	1451
Urgent	132	11	121
*Type of ICU:*			
Coronary artery disease	2808	54	2753
Congestive heart failure	1315	175	1140
Acute coronary syndrome	279	22	257
*Average heart rate (bpm)*	80	100–110	60–100
*Average respiratory rate (cpm)*	21	12–20	≤12 or ≥20
*Prescription drugs:*			
Cholesterol lowering medications	12.83%	0.14%	99.86%
ACE inhibitors	14.62%	0.38%	99.62%
Bronchodilators	10.66%	0.47%	99.53%
Diuretics	9.1%	1%	99%
Insulins	7.85%	1.04%	98.96%
Anticoagulants	7.42%	0.8%	99.2%
Electrolytes	13.22%	0.66%	99.34%
Beta blockers	7.2%	1.78%	98.22%
Antiplatelet agents and DAPT	3.46%	3.58%	96.42%
Anti-histamines	3.44%	0.53%	99.47%
Quinolone antibiotics	3.2%	1.42%	98.58%
Nitrates	2.63%	1.21%	98.79%
Peptides	1.36%	1.01%	98.99%
Glucose elevating agents	1.63%	5.88%	94.12%
Antidysrhythmics	0.89%	4.6%	95.4%
Calcium channel blockers	0.35%	3.95%	96.05%
Sulfonic acid	0.15%	6.06%	93.94%
*Hospital length of stay*	2.9 days	3.1 days	2.7 days

**Table 2 healthcare-10-00966-t002:** Parameters of models.

Decision Trees	Gradient Boosted Models
Criterion: Entropy	Number of estimators: 2000
Max depth: 10	Learning rate: 0.3
Splitter: Best	Criterion: MSE
Max features: log2	Min sample leaf: 2
Min samples leaf: 4	Min samples split: 5
Min samples split: 10	

**Table 3 healthcare-10-00966-t003:** Performance comparison summary between RM values, accuracy, and prediction time metrics.

			Linear Regression	Trees Regression				Mixed (Blenders)		
Exp 1	Model		LR *	TR1 *	TR2 *	TR3 *	TR4 *	B1 *	B2 *	B3 *
	Accuracy %		0.89	0.783	0.917	0.761	0.726	0.98	0.935	0.961
	RM	Validation (18%)	0.00682	0.18	0.00399	0.01714	0.01353	-	-	-
		Cross Validation (36%)	0.00575	0.09	0.00261	0.00953	0.01357	-	-	-
		Holdout (72%)	0.00539	-	0.00173	0.00521	0.00312	0.0012	0.00158	0.00058
	Prediction time (mS)		5627.83	3734.79	6914.43	22,443.8	17,452.56	123,450.9	123,800	69,376.8
Exp 2	Accuracy %		0.88	0.77	0.91	0.77	0.756	0.978	0.91	0.956
	RM	Validation (18%)	0.00679	0.18	0.00399	0.01714	0.01353	-	-	-
		Cross Validation (36%)	0.00571	0.09	0.00261	0.00953	0.01357	-	-	-
		Holdout (80%)	0.00512	-	0.00159	0.00503	0.00298	0.0012	0.00147	0.00054
	Prediction time (mS)		5628.01	3699.45	6999	23,166.4	20,514.6	123,551	132,500	70,015.2
Exp 3	Accuracy %		0.89	0.78	0.914	0.76	0.754	0.937	0.914	0.915
	RM	Validation (18%)	0.00659	0.18	0.00373	0.01694	0.0112	-	-	-
		Cross Validation (46%)	0.00551	0.085	0.00191	0.00958	0.01057	-	-	-
		Holdout (72%)	0.00512	-	0.00159	0.00503	0.00298	0.80319	0.00147	0.00054
	Prediction (mS)		5568	4697	7859	23,894.2	25,735.02	215,693	151,236	78,020
Exp 4	Accuracy %		0.89	0.781	0.925	0.709	0.781	0.95	0.965	0.89
	RM	Validation (25%)	0.00614	0.1	0.00329	0.01658	0.01123	-	-	-
		Cross Validation (36%)	0.00541	0.09	0.00261	0.00953	0.01357	-	-	-
		Holdout (72%)	0.00481	-	0.00148	0.00493	0.00298	0.000001	0.9947	0.00054
	Run time for 100 Predictions (mS)		5750	3610.9	6990	32,548	24,590.4	133,511	15,039.3	699,081

* LR: Auto-tuned stochastic Gradient Descent Regression * TR1: Decision Tree; * TR2: Extreme Gradient Boosted Trees Regression with Early Stopping * TR3: Gradient Boosted Greedy Trees Regression with Early Stopping * TR4: Light Gradient Boosted Trees Regressor with Early Stopping. * B1: Advanced Generalized Linear Regression Model (GLRM); * B2: Efficient Neural Network (ENET); * B3: AVG Blender.

## Data Availability

MIMIC III Dataset.

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
