# Peer review of "A Machine Learning Based Discharge Prediction of Cardiovascular Diseases Patients in Intensive Care Units"

_healthcare, 2022, doi:10.3390/healthcare10060966_

Round 1

Reviewer 1 Report

I thank the authors for this interesting work. Obviously, a considerable effort has been done in response to earlier reviews. That said, I have no major concerns in general. However, please consider the points below for the final version.

(1)

The contributions should be described more precisely in the introduction. Part of the introduction should elaborate on the contributions in line with the literature.

(2)

I find it is necessary to consider the role of Modeling and Simulation (M&S) for discharge planning. development of healthcare policies has intensively relied on M&S methods. For example, analytical questions including what-if scenarios would be properly tackled using M&S. Therefore, the present study should refer to part of the M&S contributions in this regard.

(3)

As above-mentioned, part of the introduction or related work should refer to studies that applied M&S for discharge planning systems. For example:

https://doi.org/10.1145/2901378.2901381

https://doi.org/10.1109/TASE.2019.2896271

(4)

Please mention the software libraries used, and please cite their references.

(5)

For the future work, I suggest touching further on the model explainability using frameworks such as SHAP or LIME.

Author Response

Reviewer 1, comment 1:

The contributions should be described more precisely in the introduction. Part of the introduction should elaborate on the contributions in line with the literature.

Answer: Thank you for your comment which we fine interesting in terms of giving the paper a more elaborated organization. We believe we have done necessary changes that go with what was requested.

Reviewer 1, comment 2:

I find it is necessary to consider the role of Modeling and Simulation (M&S) for discharge planning. development of healthcare policies has intensively relied on M&S methods. For example, analytical questions including what-if scenarios would be properly tackled using M&S. Therefore, the present study should refer to part of the M&S contributions in this regard.

Answer: Indeed, we have maybe misvalued the importance of such axis to mention the importance of our work. As a response to the comment, we have added a part that considers the contributions based on modeling and simulation in the field of tackling healthcare issues.

Reviewer 1, comment 3:

As above-mentioned, part of the introduction or related work should refer to studies that applied M&S for discharge planning systems. For Example :

https://doi.org/10.1145/2901378.2901381

https://doi.org/10.1109/TASE.2019.2896271

Answer: Thank you so much for the suggestions. We believe we have taken them into account.

Reviewer 1, comment 4:

Please mention the software libraries used, and please cite their references.

Answer: Indeed, this is a very interesting point to be clarified. We have added libraries and packages that have been used in building models in our work. We also have including references into their official platforms. The libraries include pre-built functions to process data and train models also to evaluate the model’s performance.

Reviewer 1, comment 5:

For the future work, I suggest touching further on the model explainability using frameworks such as SHAP or LIME.

Answer: that s a very constructive and helping comment. We will be taking this amazing point into consideration for further works of course.

Reviewer 2 Report

In this paper, the authors aim to optimize hospitals’ resources and to maximize service quality by using a number of methods to handle CVD patients’ discharge from ICUs using a dataset that contains physiological and historical data with the help of machine learning algorithms.

Discharge planning is an important issue, and unfortunately, it suffers from a lot of variability in the clinical decision-making processes.

In hospitals, most of the intensive care units do not have a written patient discharge guideline.

Therefore, as a decision support system, the proposed work is very useful. 

The manuscript is directly related to the topics/content of the Healthcare Journal. 

I have some objections about the acceptance of the paper with its current form. I have listed my reviews as follows.

One of the important things that I like about this paper is that the authors did a good data analysis before the training process.

As seen from Figure-2 a good data preprocessing has been done. 

The data is categorized depending on the features as shown in "Table 1. Baseline patient’s characteristics and outcome measures."

1) 
Authors said that "The model achieved an optimized training accuracy value of 0.98"
What does this accuracy mean?
How do you calculate it?

The authors try to measure "a patient’s readiness for discharge.". (paper entitled "A Machine Learning based Discharge Prediction of Cardiovascular Diseases Patients in Intensive Care Units")

This is a regressional value. How do you measure whether this value is accurate of not?

2)  
I want to learn how the authors divide the training and testing dataset.
What is the efferat of the randomeness here? 
Do you use a K-fold approach?

3)
"Figure 3. Feature selection of informative features." is not clear and readable. Please refresh it with some high resolution images.

Characters are to small and really hard to read.

4) Figure 4 is so confusing. There are too many colors in it. Each color has a different meaning. However, they overlap and therefore it is hard to follow the lines.

There are some unconnected lines also. 

Decision Tree Flow is shown with black arrows, but there are some additional black lines also. 

5)
The authors said that "we conducted four experiments on different data partition configurations" 
What do these partitions mean?

Table 3 is not clear in this manner.

"Accuracy % 0.89" What do you mean with this accuracy level % 0,89 of %89?
"Prediction (mS) 5568"   what do you mean with prediction time. If this is a decision process, it is too long.
"Run time for 100 Predictions (mS) 5750". How do you calculate it?
As a single data? or read each data one by one?

6)
References are not formatted uniformly. It should fit the journal's format.

Author Response

Point by point revisions and responses:

Reviewer 2, comment 1:

Authors said that "The model achieved an optimized training accuracy value of 0.98"
What does this accuracy mean?
How do you calculate it?

The authors try to measure "a patient’s readiness for discharge.". (Paper entitled "A Machine Learning based Discharge Prediction of Cardiovascular Diseases Patients in Intensive Care Units")

This is a regression value. How do you measure whether this value is accurate of not?

Answer: We appreciate your very specific and constructive comments. Indeed, here discharge prediction means predicting when with “what percentage” a patient can be “securely” discharged. Here discharge location can be home, another hospital or institution or simply a lower care unit. As for the models used, as you might already noticed we are using decision trees, and gradient regression models which all are used in prediction tasks.

Indeed, to evaluate our models’ performance, we mentioned three main parameters: Residual mean, running time and accuracy. Let’s go step by step. What are residuals? Residuals are a measure of how far from the regression line data points are. This means that residual mean will calculate the sum of mean error over the number of total sample observations. Great! Now by accuracy we want to know how close the predictions were to the expected values. To simplify the calculus for readers we have used accuracy =1-error. The error here is calculated using the residual mean of the four runs mentioned in table 2.

Reviewer 2, comment 2:

I want to learn how the authors divide the training and testing dataset.
What is the effect of the randomness here? 
Do you use a K-fold approach?

Answer: thank you for your comment. Actually, we are not using K-fold approach, instead we wanted to study the impact of data partition on the models’ performance, we conducted four experiments on different data partition configurations. The configurations and results after simulations are summarized in table 3.

Reviewer 2, comment 3:

"Figure 3. Feature selection of informative features." is not clear and readable. Please refresh it with some high-resolution images.

Characters are to small and hard to read.

Answer: Thank you, you are right indeed. We have placed the previous figures (Figure 3.a, b, c, d, and e) by new ones with high resolution.

Reviewer 2, comment 4:

Figure 4 is so confusing. There are too many colors in it. Each color has a different meaning. However, they overlap and therefore it is hard to follow the lines.

There are some unconnected lines also. 

Decision Tree Flow is shown with black arrows, but there are some additional black lines also. 

Answer:  We have tried to make the figure clearer. We hope it answers the issue.

Reviewer 2, comment 5:

The authors said that "we conducted four experiments on different data partition configurations" 
What do these partitions mean?

Table 3 is not clear in this manner.

"Accuracy % 0.89" What do you mean with this accuracy level % 0,89 of %89?
"Prediction (mS) 5568"   what do you mean with prediction time. If this is a decision process, it is too long.
"Run time for 100 Predictions (mS) 5750". How do you calculate it?
As a single data? or read each data one by one?

Answer: thank you for your comment. We would first to mention our very much appreciation for a such a detailed comment. Okay, so let’s go step by step.

The partitions mentioned in table 3 represent “how” we divided our data to train our models.

The running time (which we could calculate using the time library of python by which you can calculate the time that a specific program takes to run) here does not include the training and testing of the models but also considers the execution time of the whole program which also includes uploading the data.

Every run of these experiments is not a single iteration but 150 iterations.

Reviewer 2, comment 6:

References are not formatted uniformly. It should fit the journal's format.

Answer: Yeah, in fact, those rates represent partitions of Validation, Cross-validation, and holdout for every experiment.

We have formatted references accordingly to the given guidelines in: https://www.mdpi.com/authors/references. 

Reviewer 3 Report

I do not have more questions. The authors have now replied to all my concerns. I consider that the paper can be considered for publication.

Author Response

Reviewer 3, comment:

I do not have more questions. The authors have now replied to all my concerns. I consider that the paper can be considered for publication.

Answer: we would like to thank you tremendously because it’s due to the reviewers comments that do better work.

Reviewer 4 Report

The revisions improved the manuscript.  Why do the authors use tree-based models for prediction in the study? The motivation should be clarified. Should the values of "Accuracy %" in Table 3 be 100 times? Some essential works that use tree-based machine learning models for physiological data, such as 10.1038/s41598-020-62133-5, 10.3390/e18080285, and others, were missing, and the authors should introduce these works.

Author Response

Reviewer 4, comment1:

The revisions improved the manuscript.  Why do the authors use tree-based models for prediction in the study? Should the values of "Accuracy %" in Table 3 be 100 times?

Answer: Thank you so much for the comment. The main reason for which we choose to use tree-based models is because they are simple to understand and to interpret and performs well even if its assumptions are somewhat violated.

Every accuracy value represents the accuracy of a specific experiment and a specific model in hand.

Reviewer 4, comment 2:

Some essential works that use tree-based machine learning models for physiological data, such as 10.1038/s41598-020-62133-5, 10.3390/e18080285, and others, were missing, and the authors should introduce these works.

Answer: thank you so much for the suggestions. We have taken them into account.

Round 2

Reviewer 4 Report

The revisions improve the manuscript. It can be accepted now.

This manuscript is a resubmission of an earlier submission. The following is a list of the peer review reports and author responses from that submission.

Round 1

Reviewer 1 Report

Authors proposed cardiovascular desease prediction. The paper is well organized and written, with an intensive theoretical review. However, authors used some classical and already used models for prediction. In general, the reader can not see the novelty of the proposed approach. 

Author Response

We appreciate your comment. In fact, as we are checking the literature review, it appears that there’s no work or publication (either by the time we developed these models presented in the current paper or even by the time we are revising it) that presents such work with:

  • As many models as we present (Auto-tuned stochastic gradient descent regression, decision tree, extreme gradient boosted trees regression with early stopping, gradient boosted greedy regression with early stopping, light gradient boosted trees regressor with early stopping, advanced generalized linear regression model, efficient neural network and AVG blender)
  • We have repeated the training of models four times with randomly split data to verify if results are biased.
  • No other works have used MIMIC III for patients discharge prediction most of the work done on MIMIC III (or on previous editions) aimed to predict readmission or mortality.
  • We grouped impact of physiological signs, drug therapy but also chronological variants like time of admission and transfers in between units on the decision of a patient’s readiness to discharge which is in most cases separated.

Reviewer 2 Report

The paper is entitled "A Machine Learning based Discharge Prediction of Cardiovascular Diseases Patients in Intensive Care Units"

Authors are : Kaouter Karboub and Mohamed Tabaa

In this paper, the authors aim to optimize hospitals’ resources and to maximize service quality by using a number of methods to handle CVD patients’ discharge from ICUs using a dataset that contains physiological and historical data with the help of machine learning algorithms.

They tested their proposed model on 4226 cardiovascular disease patients, and they reached an accuracy value of 0.98.

Discharge planning is an important issue, and unfortunately, it suffers from a lot of variability in the clinical decision-making processes.

In hospitals, most of the intensive care units do not have a written patient discharge guideline.

Therefore, as a decision support system, the proposed work is very useful. 

The manuscript is directly related to the topics/content of the Healthcare Journal. 

I want to see this paper in the journal after making some small revisions, which are defined below.

One of the important things that I like about this paper is that the authors did a good data analysis before the training process.

As seen from Figure-2 a good data preprocessing has been done. 

The data is categorized depending on the features as shown in "Table 1. Baseline patient’s characteristics and outcome measures."

Firstly,

I want the authors to explain how they measure the accuracy of the system. They said that they reached about 0.98 acccuracy. What does it mean?

Because this is not a classification system. 

The authors try to measure "a patient’s readiness for discharge.". (paper entitled "A Machine Learning based Discharge Prediction of Cardiovascular Diseases Patients in Intensive Care Units")

This is a regressional value. How do you measure whether this value is accurate of not?

Secondly,

The authors said that "the data was partitioned randomly." However, for data science, this randomness generates different results for each execution. Therefore, it would be better to use a k-fold approach (i.e. 5-fold)  for making tests.

Thirdly,

"Figure 3. Feature selection of informative features." is not clear and readable. Please refresh it with some high resolution images.

What is the significance of the "Day of Month " analysis in Figure 3. Does it have a statistical effect in the analysis.

Fourthly,

Figure 4 is so confusing. There are too many colors in it. Each color has a different meaning. However, they overlap and therefore it is hard to follow the lines.

There are some unconnected lines also. 

Decision Tree Flow is shown with black arrows, but there are some additional black lines also. 

Fifthly,

There is too much similarity in the whole "2-Literature Review" section and the final paragraph of the "Discussion Section".

The authors must check these sections carefully.

Sixthly, 

The authors said that "we conducted four experiments on different data partition configurations {18%, 36%, 72%}; {18%, 36%, 80%}; {18%, 46%, 72%} and {25%, 36%, 72%}.". 
What do these partitions mean?

References are not formatted uniformly. It should fit the journal's format.

Finally,

Although the paper has a good level of English, some parts need to be revised.

I listed some of them as follows. But there are also others. 

how can we handle the crisis of scarce resources through effective discharge based on these patients’ Electronic Health Records (EHR) and using machine learning technique.--how can we handle the crisis of scarce resources through effective discharge based on these patients’ Electronic Health Records (EHR) and using machine learning technique?

Electronic Health Records (EHR)--Electronic Health Records (EHRs)

We avail results from our previously tested mathematical model-- ?? 

From the other hand, we trained multiple regression models  -- On the other hand, we trained multiple regression models

The training (75%) & validation (25%) dataset  -- The training (75%) and validation (25%) dataset

We run multiple experiments to study data partition’s  -- We ran multiple experiments to study data partition’s

evaluated model is 123450.9 mS.  -- evaluated model is 123450.9 ms.

the most influencing patterns to judge a patient’s readiness --the most influential patterns to judge a patient’s readiness 

In literature, many studies were attracted by the complexity  -- In the literature, many studies are attracted by the complexity

using this patient health status--using this patient's health status

In the other hand, a new trend--On the other hand, a new trend

predict patients that are more likely to be discharged among other patients.--predict patients who are more likely to be discharged among other patients.

From the other hand, there are relatively---On the other hand, there are relatively

and being responsible of the utilization--and being responsible for the utilization

split it into: (75%) training data and (25%) which includes--split it into: (75%) training data and (25%) test data which include

seizure& transient ischemic attack--seizure and transient ischemic attack

Author Response

Reviewer 2-comment 1:

I want the authors to explain how they measure the accuracy of the system. They said that they reached about 0.98 accuracy. What does it mean?

Because this is not a classification system. 

The authors try to measure "a patient’s readiness for discharge.". (Paper entitled "A Machine Learning based Discharge Prediction of Cardiovascular Diseases Patients in Intensive Care Units")

This is a regression value. How do you measure whether this value is accurate or not?

Response:

We appreciate your very specific and constructive comments. Indeed, here discharge prediction means predicting when with “what percentage” a patient can be “securely” discharged. Here discharge location can be home, another hospital or institution or simply a lower care unit. As for the models used, as you might already noticed we are using decision trees, and gradient regression models which all are used in prediction tasks.

Indeed, to evaluate our models’ performance, we mentioned three main parameters: Residual mean, running time and accuracy. Let’s go step by step. What are residuals? Residuals are a measure of how far from the regression line data points are. This means that residual mean will calculate the sum of mean error over the number of total sample observations. Great! Now by accuracy we want to know how close the predictions were to the expected values. To simplify the calculus for readers we have used accuracy =1-error. The error here is calculated using the residual mean of the four runs mentioned in table 2.

Reviewer 2-comment2:

The authors said that "the data was partitioned randomly." However, for data science, this randomness generates different results for each execution. Therefore, it would be better to use a k-fold approach (i.e., 5-fold) for making tests.

Response: your comment is very constructive. We will take it into consideration for further works.

Reviewer 2-comment3:

"Figure 3. Feature selection of informative features." is not clear and readable. Please refresh it with some high-resolution images.

What is the significance of the "Day of Month " analysis in Figure 3? Does it have a statistical effect in the analysis?

Response:

Thank you, you are right indeed. We have placed the previous figures (a, b, c, d, and e) by new ones with high resolution. Yes, we wanted to see how a specific period of the months can affect the flow of patients in intensive care units which can be associated to maybe logistic related factors like for example the availability of used drugs. As a result, we have taken admission time (day of the month) as an informative feature.

Reviewer 2-comment 4:

Figure 4 is so confusing. There are too many colors in it. Each color has a different meaning. However, they overlap and therefore it is hard to follow the lines.

There are some unconnected lines also. 

Decision Tree Flow is shown with black arrows, but there are some additional black lines also. 

Response:

We have tried to make the figure clearer. We hope it answers the issue.

Reviewer 2-comment 5:

There is too much similarity in the whole "2-Literature Review" section and the final paragraph of the "Discussion Section".

The authors must check these sections carefully.

Response:

Thank you so much for this comment, it is indeed very constructive. We have done necessary changes.

Reviewer 2-comment 6:

The authors said that "we conducted four experiments on different data partition configurations {18%, 36%, 72%}; {18%, 36%, 80%}; {18%, 46%, 72%} and {25%, 36%, 72%}.". 
What do these partitions mean?

References are not formatted uniformly. It should fit the journal's format.

Response:

Yeah, in fact, those rates represent partitions of Validation, Cross-validation, and holdout for every experiment.

We have formatted references accordingly to the given guidelines in https://www.mdpi.com/authors/references. 

Reviewer 2-comment 7:

Although the paper has a good level of English, some parts need to be revised.

I listed some of them as follows. But there are also others. 

how can we handle the crisis of scarce resources through effective discharge based on these patients’ Electronic Health Records (EHR) and using machine learning technique? --how can we handle the crisis of scarce resources through effective discharge based on these patients’ Electronic Health Records (EHR) and using machine learning technique?

Electronic Health Records (EHR)--Electronic Health Records (EHRs)

We avail results from our previously tested mathematical model-- ?? 

From the other hand, we trained multiple regression models  -- On the other hand, we trained multiple regression models

The training (75%) & validation (25%) dataset  -- The training (75%) and validation (25%) dataset

We run multiple experiments to study data partition’s  -- We ran multiple experiments to study data partition’s

evaluated model is 123450.9 mS.  -- evaluated model is 123450.9 ms.

the most influencing patterns to judge a patient’s readiness --the most influential patterns to judge a patient’s readiness 

In literature, many studies were attracted by the complexity  -- In the literature, many studies are attracted by the complexity

using this patient health status--using this patient's health status

In the other hand, a new trend--On the other hand, a new trend

predict patients that are more likely to be discharged among other patients.--predict patients who are more likely to be discharged among other patients.

From the other hand, there are relatively---On the other hand, there are relatively

and being responsible of the utilization--and being responsible for the utilization

split it into: (75%) training data and (25%) which includes--split it into: (75%) training data and (25%) test data which include

seizure& transient ischemic attack--seizure and transient ischemic attack

Response:

Thank you, we hope the changes we have done meet the necessary expectations.

Reviewer 3 Report

In this paper, the authors evaluate multiple machine learning models to predict when a patient admitted to ICUs can be transferred to a lower care unit or sent home. The models were build and tested using the MIMIC III dataset.

After revising the manuscript, I consider that explanations must be revised and improved a lot. It resulted difficult to read the paper and understand the proposal.

I recommend to reject the paper.

Please, see my comments in the attached pdf file.

Author Response

Thank you for your comment. We have tried to improve the paper’s clarity and to revise the paper to make it more readable and give it more insights. We hope the changes meet your expectations.

Reviewer 4 Report

The presented contents are very hard to follow due to grammar mistakes, vague figures, and illogical organization, which makes it impossible to evaluate the novelty/scientific value/significance of the work.  

Author Response

Your comment is interesting, we hoped that you made your comment clearer with specific points to clarify but in all cases, we have tried to improve the paper, evaluate the English, we re-simulated the models to replace figures with others that have higher resolution.

In the other hand, we hope that you consider the fact that models used in this paper are not used in any other works with such an interesting database and such enormous processing

Reviewer 5 Report

1. Most ML algorithms except for the mixed or ensemble algorithms in this study are tree-based. Can the "machine learning based" in the title of this paper be changed to "tree-based"?
2. How is the "Tree based algorithm" in Fig. 4 used for preprocessing? More details are needed.
3. Please explain the "data partition configurations" in L371-372. 
4. "RESIDUAL MEAN" or "Residual Mean". Please use uniform spelling.
5. To show the performance of ML on disease classification and/or prediction, the authors are encouraged to introduce and cite some related works in Introduction, such as 10.1038/s41598-020-62133-5, 10.3390/e18080285, 10.1002/9781119769262.ch8, and others.

Author Response

First, we would like to thank for the very constructive and highly organized and concise comment.

Let’s go then point by point in the same order you mentioned in the comment below:

  1. Would you mind me if I ask, aren’t decision trees a type of supervised machine learning? We think that in both cases (the article being titled “machine learning based” or “tree based”), it won’t change the content of the article.
  2. Tree based here is used to impute data that are missing from categorical data.
  3. Yeah, in fact, those rates represent partitions of Validation, Cross-validation, and holdout for every experiment.
  4. We have done necessary changes.
  5. Thank you for the suggestions.

Round 2

Reviewer 3 Report

After a detailed review, I would like to point that the authors did not address any of my comments. They didn’t reply point by point to my concerns, neither make any change in the text related to them.

I consider that the changes made were minor. The work is very difficult to understand, it still needs to improve a lot.

I do not consider that another major revision like this would be enough.

Reviewer 5 Report

The revisions are mainly reflected in the text format, and they don't improve the paper clearly. Many tree-based machine learning algorithms were used in the study, but their roles and parameter settings are not clear. The current Figure 4 is hard to read and detailed explanations are required.